# Emotional Brain Network Community Division Study Based on an Improved Immunogenetic Algorithm

**DOI:** 10.3390/brainsci12091159

**Published:** 2022-08-30

**Authors:** Renjie Zhao, Tao Zhang, Shichao Zhou, Liya Huang

**Affiliations:** 1Bell Honors School, Nanjing University of Posts and Telecommunications, Nanjing 210023, China; 2School of Materials Science and Engineering, Nanjing University of Posts and Telecommunications, Nanjing 210023, China; 3School of Computer Science, Nanjing University of Posts and Telecommunications, Nanjing 210023, China

**Keywords:** emotion analysis, community division, genetic algorithm, immunity operator, emotional brain functional network

## Abstract

Emotion analysis has emerged as one of the most prominent study areas in the field of Brain Computer Interface (BCI) due to the critical role that the human brain plays in the creation of human emotions. In this study, a Multi-objective Immunogenetic Community Division Algorithm Based on Memetic Framework (MFMICD) was suggested to study different emotions from the perspective of brain networks. To improve convergence and accuracy, MFMICD incorporates the unique immunity operator based on the traditional genetic algorithm and combines it with the taboo search algorithm. Based on this approach, we examined how the structure of people’s brain networks alters in response to different emotions using the electroencephalographic emotion database. The findings revealed that, in positive emotional states, more brain regions are engaged in emotion dominance, the information exchange between local modules is more frequent, and various emotions cause more varied patterns of brain area interactions than in negative brain states. A brief analysis of the connections between different emotions and brain regions shows that MFMICD is reliable in dividing emotional brain functional networks into communities.

## 1. Introduction

Emotions are the expression of human attitudes toward objective items, and they play an important role in decision-making, cognition, creativity, and interpersonal communication [1]. Depending on people’s comprehension and evaluation of the situation, they may respond emotionally differently to stimuli that are emotionally evocative [2,3,4,5]. Therefore, it is critical to comprehend how human cognitive processes and emotions interact in a particular scenario. Due to the general variation in cultures and nations, emotion classification based on physiological signal features (mostly comprising peripheral nervous system and brain signals) will produce more accurate findings than emotion classification based on non-physiological signal aspects (voice, facial expressions, posture, etc.) [6,7,8].

Electroencephalography (EEG) [9,10,11] is a key tool for measuring the complex signals produced by neural activity that were generated by various emotions. It provides realistic brain mapping of spatial and temporal information about brain function [12], making it one of the most popular techniques used to study the brain mechanisms underlying emotional arousal processes.

The dynamic changes in brain structure that occur during various emotions are not well understood by researchers because the brain areas are a complex dynamic system. A graphical technique is taken into consideration to thoroughly study the patterns of functional connectivity in the brain since the channel information gathered by discrete electrodes in EEG devices is insufficient to depict the coupling connections between brain areas. Researchers are increasingly concentrating on the study of the functional connectivity of the brain based on complex networks, as the idea of “human connectivity” has gained popularity. They are attempting to use the variability of the properties of complex networks to describe changes in the state of the brain.

The brain contains specific areas that are linked to emotional activity [13], and EEG-based connectivity analysis has allowed researchers to better understand how the human brain processes emotions. Additionally, it is believed that functional brain networks represent the intrinsic activity of the brain, which may provide important details regarding the communication between various brain areas [14,15,16,17]. Different emotional states may correspond to different networks created from the EEG signals gathered during emotion evocation, which is useful for researchers to study the information exchange mechanisms of the brain during the processing of emotional activities and the variations in brain interaction patterns under different emotions. For the purpose of classifying emotions, it is necessary to employ the complex network theory to evaluate the topological characteristics of functional brain networks, which vary when emotional activity is generated.

As research has developed, it has discovered that the human brain is made up of several brain areas with linked neurons operating together. Therefore, the application of complex network theory to the division of brain network activities has gained more attention [18,19,20]. Van Den Heuvel et al. [21] used the Normalized Cut (Ncut) method to divide communities. The accuracy of the results of the Ncut algorithm depends mainly on the set graph complexity threshold and the number of group clusters, and the average accuracy is low. Adjacency Matrix Decomposition (AMD) approach was used by She et al. [22] to divide brain networks. The AMD algorithm performs better in terms of classification performance recognition results; however, the algorithm’s stability is weak. Wang et al. [23] used the Hierarchical Clustering method, which achieves better results in terms of stability and accuracy. Although these methods produce better results in the functional division of brain networks, they all require the input of certain parameter values in advance or during the process, adjusting the results of community division to achieve the best results, and do not achieve true unsupervised division. Furthermore, because the true functional division of brain networks is unpredictable, relying on artificial input parameters for division might have a detrimental influence on the outcomes.

To solve the problems mentioned above, scientists suggested the use automatic parameter input-free adaptive algorithms. Girvan Newman (GN) is a community division algorithm proposed by Newman in 2002 with the criterion of calculating the number of edge meshes [24]; Faster Newman (FN) is a fastened module division method based on hierarchical clustering [25]; Non-negative Matrix Factorization (NMF) is a dividing algorithm using non-negative matrix decomposition [26]. Modularity (*Q*) and Normalized Mutual Information (NMI) [27] values are the metrics we use to gauge the effectiveness of the network community division. *Q* measures the strength of the structure inside each community in the network, while NMI assesses the degree of similarity between the communities produced by the algorithm and the actual community division. Based on the above algorithmic ideas, this paper proposes a Multi-objective Immunogenetic Community Division Algorithm Based on Memetic Framework (MFMICD) for the community division of complex networks, which does not require the input of the number of communities and other parameter values in the network in advance and uses the values of *Q* and NMI as the objective function to obtain the best community classification possible. The main aims are as follows:Based on the traditional genetic algorithm, the specific immunity operator is introduced to improve the accuracy and accelerate the convergence of the algorithm, and the results are further optimized with a forbidden search algorithm;The adaptive threshold setting method is proposed to allow the threshold set by the binarization network to be dynamically altered in accordance with the different brain network features to increase the accuracy of the outcomes of brain division;*Q* and NMI metrics derived by the techniques are reliable and superior to the currently used brain network division algorithms under the same experimental settings;The algorithm is used to divide the brain network into communities in various emotional states and investigate changes in how the brain communicates across regions and interaction patterns when processing emotional activity, yielding results that are largely consistent with previous physiological findings.

The remainder of this paper is organized as follows. The system model of the MFMICD and data processing flow are described in Section 2. The results and discussion of the structural division of brain networks are provided in Section 3. Conclusions are drawn in Section 4.

## 2. Method

### 2.1. Algorithm Framework

In this study, a Multi-objective Immunogenetic Community Division Algorithm Based on Memetic Framework (MFMICD) was developed using immune network theory, which analyzes different emotional states of individuals by calculating the importance of nodes in the brain network and divides the brain functional network into communities based on the objective function values. The algorithm framework is shown in Figure 1. The following are a sequence of algorithmic stages: (a) Data pre-processing. The data is filtered out for experiment after the time series segments in the steady state in the data set are intercepted. (b) Complex network construction based on phase-locked values (PLV). PLV statistical techniques are used to extract connection metrics and produce PLV matrices. (c) Matrix binarization. Set a binarization threshold to create an undirected binarized brain network. (d) Brain function network establishment. We assume that linked edges exist between the related nodes when an element’s value in the binarization matrix is 1, and vice versa. (e) Brain network structure division. The MFMICD algorithm separates the brain functional network into several functional communities using the obtained brain functional network matrix as its input.

### 2.2. Data Pre-Processing

#### 2.2.1. EEG Emotional Database (DEAP)

We used the Database for Emotion Analysis using Physiological Signals (DEAP) [28], which records physiological signals emitted by 32 different subjects while watching 40 different types of music videos (60 s each) based on EEG signals of different emotions induced by music videos. The 32 participants in the experiment, including 16 men and 16 women, ranged in age from 19 to 37 years old, with an average age of 26.9 years. The specific process of EEG acquisition includes the following steps:

(1) Baseline Recording Acquisition: The participant will endeavor to maintain composure and record the marker for the beginning of the EEG signal while the operation lasts for 5 s and a cross is displayed on the screen.

(2) Music Video Play: This process will last 63 s, and the first 3 s are for each video conversion and the last 60 s are for the music video to actually play.

(3) Self-assessment Scoring: Participants were asked to self-assess on the SAM questionnaire after each video was shown, grading each video based on Valence, Arousal, Dominance, and Liking, which are used to analyze physiological signals in multimodality.

A Biosemi ActiveTwo system with 512 Hz sampling was employed as the EEG acquisition device for the DEAP database. The experiment was conducted using presentation software provided by Neurobehavioral to play music videos on a Pentium four computer with a monitor size of 1280 × 1024. Although the device is capable of recording 128 lead EEG data, it only used 32 electrode caps during the acquisition. The electrode placement distribution essentially covers the four primary brain areas when 32 electrode caps are inserted following the international 10–20 system, as indicated in the blue circle in Figure 2.

#### 2.2.2. Pre-Processing Methods

The DEAP dataset was preprocessed with the EEGLAB tool in MATLAB. Because the connectivity pattern of the subjects’ functional brain networks was difficult to keep relatively stable at the start and end of the data collection phase, the 60 s time series was first divided into three equal parts of 20 s each before constructing the network, and the middle one (21 s, 40 s) was finally chosen as the time series for calculation. After doing multi-band studies and referring to the results of existing studies [29], we discovered that the phase coupling in the *θ*-band strongly correlates with changes in mood; hence, the *θ*-band data with frequencies of (4 Hz, 7 Hz) were used for experiments to confirm the correctness of the experimental results.

### 2.3. Network Feature Extraction

#### 2.3.1. PLV Statistical Analysis

The connection metric is extracted using the PLV statistical approach [30], and PLV may be utilized to examine instantaneous changes in connectivity. The PLV of two continuous-time signals $x_{i}(t)$ and $x_{j}(t)$ is as stated in Equation (1):(1)$$PLV=\frac{1}{N}\left|\sum_{n=1}^{N}\exp(i\theta (t))\right|$$
where *N* is the total number of samples in the period; θ(t) represents the phase difference between $x_{i}(t)$ and $x_{j}(t)$ at moment *t*. The PLV matrix values that are obtained in a range between 0 and 1, with 0 denoting desynchronization and 1 synchronization.

#### 2.3.2. Matrix Binarization

The PLV matrix has been binarized to allow for more efficient data calculation and algorithm iteration. A suitable binarization approach must be chosen for various research settings since the number of edge queues in the network directly influences the outcome of the community division. Hurtado et al. [31] employed the alternative data approach to identify thresholds that led to the generation of new data with the same power spectrum as the original data but without its linear features. Using EEG data, Langer et al. [32] automatically built functional brain networks based on the spatial correlations of the grid’s vertices.

However, we discovered that the findings generated by the current thresholding approaches did not clearly identify the brain network topology in various emotional states when the aforementioned method was applied to the DEAP dataset. We saw that several of the binary matrices looked to be all zeros or all ones, which would be quite incorrect for subsequent network property extraction.

Using the adaptive threshold setting strategy suggested in Section 3.1 below, we carried out further trials to improve the accuracy of the brain network classification findings. The thresholds under the adaptive thresholding method can change dynamically depending on the sensitivity of various participants, making it easy to analyze the functional brain structure under various emotional states.

### 2.4. Connectivity Analysis

#### 2.4.1. Brain Function Network Establishment

The establishment of the brain’s functional network is completed by mapping the binarized adjacency matrix to the brain’s network, with element values of 1 denoting the presence of connected edges between adjacent nodes and 0 denoting the lack of connected edges between adjacent nodes.

#### 2.4.2. Brain Network Structure Division

In this paper, optimization is carried out on the basis of the traditional genetic algorithm, and the immune network theory is used to develop a Multi-objective Immunogenetic Community Division Algorithm Based on Memetic Framework (MFMICD).

The Memetic framework divides the process of finding the optimal solution into two stages. The best individuals with the highest fitness values are chosen in the global search phase using an adaptive multi-objective genetic algorithm. The local search phase employs a forbidden search algorithm, and the best individuals are then subjected to a limited number of mutation operations to ultimately find the local optimal solution. The algorithm flow is shown in Figure 3.

(1). Global Search

In this phase, the model employs an immunogenetic algorithm based on feature data. The crossover and mutation process of the genetic arithmetic forms the basis of genetic algorithms. We suggest the “Strongest First” crossover criterion for the crossover operation, identify the crossover target based on the fitness function value, and present a power function model to achieve dynamic adjustment of the crossover and mutation probabilities throughout the algorithm iteration. The specific steps of the algorithm are shown below.

Step 1. Code Initialization

The network is consistently encoded by the model using a real number format. Assume network *E* consists of *N* total nodes. The chromosome may be represented as a string of length *N* if each node is abstracted as a community.

Step 2. Population Initialization

The identifier (ID) passing approach is utilized in this study [30]. For example, if a random node *j* in the network has *k* neighbor nodes, its neighbor node identification may be expressed as $\Omega (i)=\{x^{1},x^{2},\cdot \cdot \cdot x^{k}\}$.

Step 3. Calculation of Fitness Function

The local optimization function is calibrated and mapped to the fitness function, with the fitness value defined to be non-negative to more directly tie the individual merit of the population to the fitness function. The method in this work employs *Q* as the objective function, and the amount of modularity directly represents the population classification’s superiority. The local optimization of the objective function is linearly calibrated to the fitness function Fit(x) to represent the merit of individuals in the population more directly. The calibration is shown in Equation (2):(2)$$Fit(x)=\left\{ \begin{array}{l} f(x)-f_{\min}(x),x<0\\ 0,x\geq 0 \end{array}\right.$$
where f(x) is the objective function, $f_{\min}(x)$ is the minimum value of the function.

Step 4. Crossover and Variation

The model suggests the “Strongest First” crossover criterion in the crossover operation, defines the crossover object by the magnitude of the fitness function value, and employs circular crossover to produce new offspring to increase the convergence accuracy of the method. To preserve population variety and avoid the objective function becoming convergent too early, high variation probability is required at the beginning of the iteration. However, as the iteration goes on, the objective function approaches the optimal solution, at which point the variation probability needs to be suitably adjusted to prevent deterioration due to excessive variation. This paper introduces a power function model and regresses the function by making use of the statistical indicators of the fitness function in order to achieve the dynamic adjustment of the crossover and variance probabilities.

The crossover and variance probabilities are all real numbers between 0 and 1, and the crossover and variance probabilities are compatible and complementary, which is quite close to the inverse function’s trend. As a result, the work is as stated in Equation (3):(3)$$f(x)=a{\left(x+m\right)}^{k}$$

The function fulfills the trend of the crossover and variation probabilities for a=± 1, m=1, and k=−1. To accomplish dynamic modification of the crossover variation probability, the function-independent variables must be coupled to the fitness function. In this study, we define φ as the fitness function’s rate of change, with fit(x) as the fitness function and f(x) as the goal function. The fitness function’s rate of change is as stated in Equation (4):(4)$$\varphi =\frac{f_{avg}}{f_{\max}-f_{\min}}$$

Putting φ into Equation (3) yields the crossover probability $P_{c}$ and variation probability $P_{m}$. The two probability indicators are constantly updated when the fitness function changes during the iteration process, bringing the algorithm closer to the true genetic process and drastically reducing the number of iterations.

Step 5. Selection Strategy

Roulette Wheel Selection (RWS) [33] is used as a selection method, which relates the probability of the selection of an individual to the ratio of the current individual’s adaptation value to the total adaptation value. High ratios indicated a high probability that the current individual would be selected. The population size is assumed to *N*, and the fitness of individual *i* is $f_{i}$. The likelihood of *i* being selected is calculated as stated in Equation (5):(5)$$P_{i}=\frac{f_{i}}{\sum_{j=1}^{N}f_{j}}$$

To select mating individuals, multiple rounds of selection are required after calculating the number of selection probabilities for each individual in the population. Each round generates a uniform random number between [0, 1], which is used as a selection pointer to determine the selected individuals. Pairing can be performed after individual selection to make crossover mutation procedures easier.

(2). Local Search

The taboo search method [34], which is a meta-heuristic optimization technique, is used in the local search phase to prevent a circular search by replicating the human memory function and creating a taboo table to block the previously explored region. Moreover, some good states in the prohibited zone are freed to maintain the search’s variety.

The complicated treatment of constraints in conventional heuristics can be avoided when solving an optimization problem with constraints by adding the constraint violation as a penalty component to the evaluation function. The fundamental form of an evaluation function with a penalty component is shown in Equation (6):(6)$$F(x)=f(x)+\lambda \times \sum_{j=1}^{m}P(R_{j},x)$$
where $P(R_{j},x)$ stands for the constraint penalty value of *j*, which is 0 if the solution meets the constraint. The likelihood of discovering a superior global optimum solution is increased by the search process’ capacity to tolerate suboptimal answers and expand beyond the local optimal solution to other parts of the solution space.

### 2.5. Evaluation Function

In this study, two evaluation functions, Normalized Mutual Information (NMI) and Modularity (*Q*), were used to verify the validity of the network division findings.

#### 2.5.1. Normalized Mutual Information

Normalized Mutual Information (NMI) is a metric for comparing algorithm classification results to real-world outcomes.

The NMI between two divisions of the network *G* produce $p_{1}$ and $p_{2}$ is shown in Equation (7):(7)$$NMI(p_{1},p_{2})=\frac{-2\sum_{i=1}^{c_{p_{1}}}\sum_{j=1}^{c_{p_{2}}}C_{ij}\log(C_{ij}N/C_{i}C_{j})}{\sum_{i=1}^{c_{p_{1}}}C_{i}\log(C_{i}/N)+\sum_{j=1}^{c_{p_{2}}}C_{j}\log(C_{j}/N)}$$
where $C_{p1}$ and $C_{p2}$ are the number of communities in the divisions of $p_{1}$ and $p_{2}$, respectively; $C_{i}$ and $C_{j}$ are the sum of elements in *C*’s row *i* and column *j*, respectively; *N* is the total number of nodes in the network. High NMI scores suggest that $p_{1}$ and $p_{2}$ are comparable. When $p_{1}$ is the network’s actual division and $p_{2}$ is the algorithm’s division, a high degree of similarity indicates good algorithm outcomes.

#### 2.5.2. Modularity

Newman [35] introduced modularity as a metric for determining the tightness of a community based on the idea that a community is made up of a sequence of closely linked nodes, which is calculated as shown in Equation (8).
(8)$$Q=\sum_{i}^{k}\left(e_{ii}-a_{i}^{2}\right)=\sum_{i}^{k}e_{ii}-\sum_{i}^{k}a_{i}^{2}=Tre-\left\|e^{2}\right\|$$
where the network is divided into *k* communities, *e* is a k×k matrix, and $e_{ij}$ is the 1/2 ratio of the number of edges connecting the nodes in community *i* and *j* to the total number of edges in the network; therefore, the overall edge ratio is $e_{ij}+e_{ji}$. In particular, $e_{ii}$ represents the ratio of the number of edges in the community *i* to the total number of edges in the network; $\sum_{i}^{k}e_{ii}$ represents the ratio of the number of edges in all communities to the total number of edges in the network, with a maximum value of 1 (single community only); $a_{i}$ represents the ratio of the number of all edge endpoints connected to the community *i* to the total number of edge endpoints in the network, i.e., $a_{i}=\sum_{j}^{k}e_{ij}$. The expected value of the ratio of the number of edges inside community *i* to the total number of edges in the network is obtained as $a_{i}^{2}$ by connecting these edge endpoints at random. A good community division outcome indicates good network *Q* values.

After proposing the model, we demonstrated the validity of the approach in the previous subsection by applying it to the LFR benchmark network [36] as well as to real-world networks [37,38,39] and achieving high values of *Q* and NMI. The technique is then used to the community division of brain networks to analyze the dynamic changes in the brain utilizing complicated network property differences.

## 3. Results and Discussion

The parameters of the algorithm utilized in the experiment are set as follows to lower the complexity of the method for better population initialization: The population size *N* is set to 100, the immunogenetic algorithm’s maximum number of iterations $max\_gen_{1}$ is set to 50, the contraindicated search algorithm’s maximum number of iterations $max\_gen_{2}$ is set to 20, the length of the contraindicated table *T* is set to 10, and the number of single iterations is set to 100.

### 3.1. Threshold Determination

When *Q* > 0.3, the complex network is said to have a community division structure, and varied threshold choices have a greater impact on the final division outcomes and evaluation function values. Due to the large differences in brain networks between individuals, traditional numerical thresholds are mapped to the number of edges in the unweighted network for normalized analysis, which preserves the unique structure where varied threshold choices greatly impact the final.

To show the experimental results more clearly, individuals with subject numbers 2 and 5 were selected from the data set to display the relationship between *Q* and the set threshold, and their emotional states were classified by valance in 40 independent experiments to obtain the relationship between *Q* and the number of edges in the unweighted network for the same individual in positive and negative emotions. The experimental results are shown in Figure 4.

Individuals experiencing negative emotions demonstrated lower *Q* values than individuals experiencing positive emotions, as shown in Figure 4, indicating that community strength is weak when individuals are experiencing negative emotions. However, the *Q* values of negative emotions were not significantly different for negative and positive emotions, indicating that *Q* alone does not distinguish the emotional state that individuals are experiencing. Furthermore, the brain function network demonstrated a certain community division structure, and it showed the same trend of change at a threshold value of less than 100, meaning that the value of *Q* decreases continuously with an increase in the number of edges; however, the degree of fluctuation of the change is not consistent due to the variability of the network structure and emotional state of different human brains.

This study also used NMI to assess the similarity of the outlined community structures, using values ranging from 0 to 1, with values closer to 1 indicating a high degree of similarity. Figure 5 shows the link between the NMI values and the number of edges in the unweighted network.

In the same pattern as *Q*, the NMI value of individuals experiencing negative emotions is lower than the NMI value of individuals experiencing positive emotions, indicating that community similarity is higher when people are experiencing positive emotions. The overall trend of NMI reduces as the threshold value rises; however, the degree of the variation of NMI values rises, showing that the absence of a substantial difference in modularity across brain function networks in various states is not critical and that the difference in community structure is important.

The threshold range of (30, 40) was set based on the measures of *Q* and NMI to obtain more accurate division results, thus modularizing the functional brain network analysis and investigating the role of different brain network modules generated in the whole brain information flow.

### 3.2. Division of Brain Functional Networks in Different Emotional States

Brain networks have a complicated structure with varying topological properties. The MFMICD method may be used to (1) investigate the brain’s dynamic changes based on complex network property differences, (2) assess the brain’s information interaction patterns, and (3) quantify the modular structure of functioning brain networks.

The MFMICD was compared to Ncut and Genetic Algorithm Cut (GAcut) algorithms on brain networks in the dataset, where the experimental participants’ EEG data came from individuals with participants 1 through 16 watching, and the algorithm was run 50 times independently, with *Q* and NMI averaged and the relationship plotted as shown in Figure 6.

The MFMICD algorithm surpasses the other two algorithms in both *Q* and NMI metrics, and the algorithm is resilient and may be employed for later brain functional network division, according to the experimental data. To target and analyze the differences in brain networks of the same individual under various emotions, the brain network of the individual with the smallest difference in arousal, S2, was chosen [40]. This was based on the premise that the difference between the lowest pleasantness and the highest pleasantness was the largest among the 40 experiments of an individual. In this study, the best *θ*-band signal for the brain network division was isolated for the tests, and Figure 7 and Figure 8 illustrate the fully linked matrix derived from PLV and the topology diagram of the binary brain network.

Figure 7 and Figure 8 show a relatively predictable pattern in the PLV-based fully linked matrix, with individuals demonstrating higher PLV metric values in the negative state than in the positive state. This indicates that the strength of the brain network connections is greater in the negative state and that the synchronization between different brain regions is higher compared to the positive state [30,41]. The finding indicates that there were significant differences in the topology of the brain networks of individuals across emotions, consistent with normal distribution and homogeneity of variance (paired *T*-test: *p* < 0.05). Negative and positive emotions have different topologies of brain networks under the same threshold, with negative emotions having tighter network connections, more weighting of linked edges, and closer information exchange.

The findings of the experiments revealed that the brain network under positive and negative emotions may be split into eight and seven communities, respectively, with communities ranging in size from one to four brain regions. Yang et al. [42] used baryonic data for a brain functional network division based on Independent Component Analysis (ICA) and found similar results. Information interactions will be generated among the nodes in the brain network during emotional activities, and the roles played by nodes in the structure of communities display their own postures, and the functional structure of the brain network under different emotions shows communalities.

To investigate the key brain regions associated with emotional activity, the modular brain network structure of S2 experiencing negative and positive emotions were mapped as shown in Figure 9. The electrodes of brain regions with the same color belong to the same functional module, and the number of color categories corresponds to the number of communities in the network. To facilitate the display of the topology of the network, we specify that a connected edge exists between two nodes when the correlation between them is higher than a set threshold. We align the color of the connecting edges within each module with the node color, and the blue line (left panel) and the purple line (right panel) represent the connections between different modules under the two emotions, respectively. As can be seen from the figure below, the brain is divided into eight modules in the positive state and seven modules in the negative state. We calculated the average number of nodes contained in each module in the positive and negative states of the brain separately. The positive state corresponds to four, while the negative state corresponds to five. It is thus obtained that the brain contains more nodes in each module in the negative state than in the positive state. What is more, the number of blue lines (five) in the left panel is clearly higher than the number of purple lines (three) in the right panel. It shows that the connections between different functional modules were stronger under positive emotions. The results suggest that the functional blocks of the human brain are more distributed under positive emotions, whereas the functional blocks of the human brain are more concentrated and demonstrated a high ability.

Further examination of Figure 9 revealed that the sub-nodes within the two largest (maximum number of nodes) modules of the modular brain network under positive and negative emotions are nearly identical: *FP1*, *AF3*, *Oz*, *C4*, *CP6*, *PO4*, *F3*, *C3*, *T7*, *AF4*, *Fz*, *FC6*, *CP2*. The findings indicate that these nodes play a key role in the human brain’s emotion generation process. These 13 common nodes correspond to four separate brain areas, according to the underlying structure of human brain division (frontal lobe, temporal lobe, occipital lobe, and central region). This finding suggests that each brain region does not work independently when the brain is involved in emotional processing. Rather, the four brain regions work together, and each brain region contains its own, which are responsible for the network architecture and synergistic work among the functional regions. The lack of a common node in the parietal area in this investigation suggests that the parietal region does not influence mood alterations, which is in line with physiological findings [43]. The parietal lobe is located in the center of the human brain, on the posterior side of the central dorsal plough sulcus, and it is responsible for helping humans grasp spatial relationships, as well as pain and touch. It does not play a significant role in mood variations.

To investigate the variability of inducing factors for various emotions, the third major module was extracted on the basis of the two largest modules. The proportion of nodes belonging to each brain region in this module to the total number of nodes was analyzed, with the statistical results shown in Table 1.

The nodes belonging to the frontal lobe and central region in the synchronized brain network accounted for the largest proportion of the summarization points of its brain regions under different emotional states, indicating that the frontal lobe and central region play a role in different emotional states, according to the analysis in the above table. Furthermore, compared to positive emotions, the number of nodes included in negative emotions was noticeably larger, and its proportion was higher in the frontal lobe, the occipital lobe, and central regions. This shows that, when experiencing positive emotions, more brain regions are engaged in emotional dominance and information connections between local modules occur more frequently.

The results of the modularity analysis of the brain network structure revealed that the topology of brain networks differed slightly depending on the emotion, with positive emotions demonstrating more non-modular characteristics and negative emotions demonstrating more concentrated brain consciousness and functional structure. Diverse emotions cause different patterns of brain area interaction, and the frontal and central regions are critical hubs for emotion management; therefore, studying and analyzing these regions is crucial in the field of individual emotion expression.

## 4. Conclusions

In this paper, we propose a Multi-objective Immunogenetic Community Division Algorithm Based on Memetic Framework (MFMICD) for the community division of complex networks, without pre-inputting the number of communities or other network parameter values and using the values of *Q* and NMI as the objective function to achieve the best community division results. The algorithm was eventually applied to brain networks, and the brain networks of people experiencing various emotions were separated into communities, revealing that diverse community structures existed in the brain. The findings suggest that the brain is involved in emotion processing through the coordinated actions of four brain regions, each with its own set of core nodes, and that the brain network connections are stronger and the synchronization between different brain regions is higher during negative states than during positive states. More brain regions are involved in emotion dominance, information interactions between local modules are more frequent, and different emotions induce different patterns of brain area interactions for positive emotions than for negative emotions. The frontal lobe and central region are important hubs for emotion control, and the connections between different emotions and brain areas are briefly analyzed, indicating that the MFMICD algorithm is feasible for the community division of brain functions. We want to use the MFMICD method in a variety of areas in the future, including emotion identification and EEG feature extraction.

## Figures and Tables

**Figure 1 brainsci-12-01159-f001:**
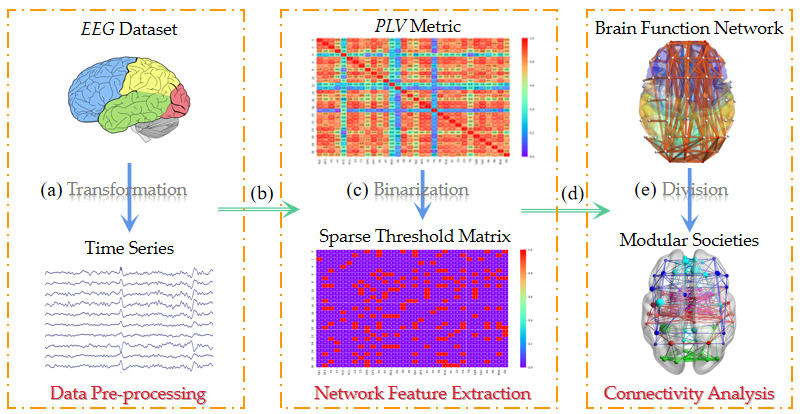
Brain network community division flow. The algorithm steps are shown in (**a**–**e**): (**a**) Acquisition and pre-processing of EEG data. (**b**) Calculate phase-locked values to construct the complex network. (**c**) Set the threshold to generate the binarization matrix. (**d**) Build the functional brain network based on binarization matrix element values. (**e**) MFMICD algorithm-based functional brain network division.

**Figure 2 brainsci-12-01159-f002:**
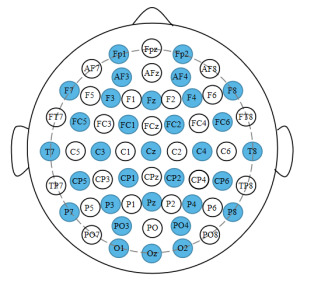
International 10–20 system with 32 electrodes (marked with blue circles).

**Figure 3 brainsci-12-01159-f003:**
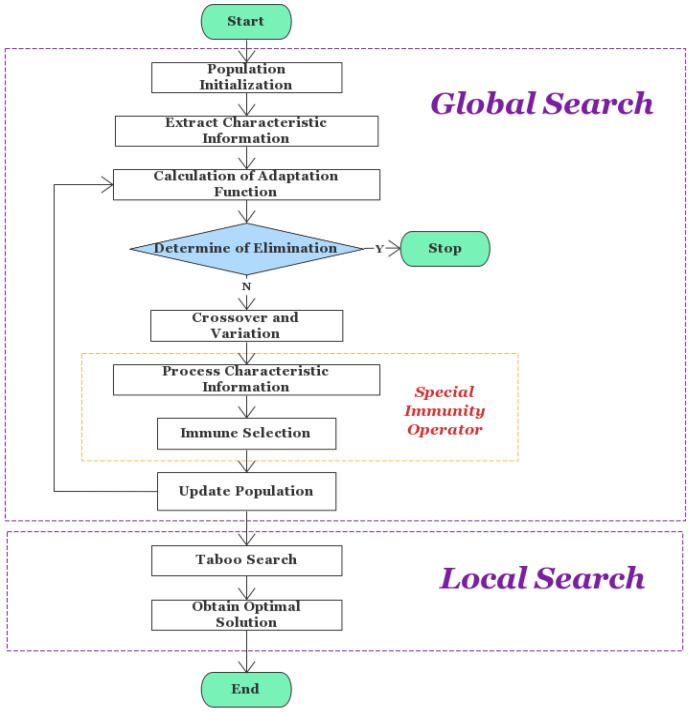
Flow chart of immune genetic algorithm based on feature information.

**Figure 4 brainsci-12-01159-f004:**
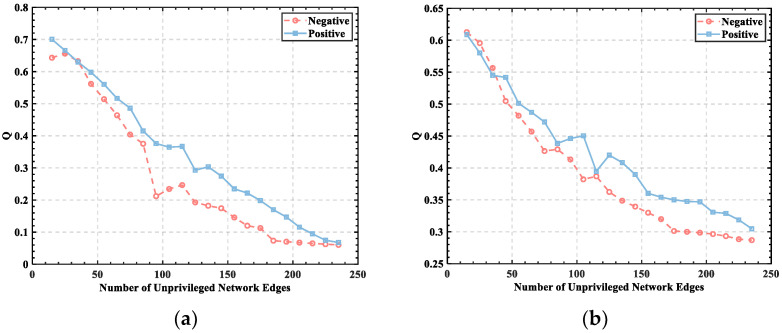
The relationship between *Q* and the number of unprivileged network edges: (**a**) S2; (**b**) S5.

**Figure 5 brainsci-12-01159-f005:**
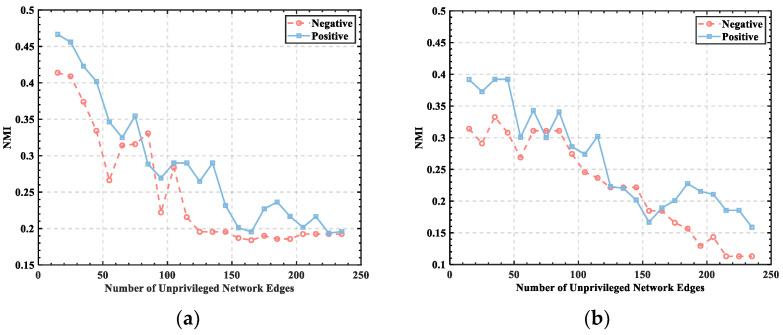
The relationship between NMI values and the number of unprivileged network edges: (**a**) S2; (**b**) S5.

**Figure 6 brainsci-12-01159-f006:**
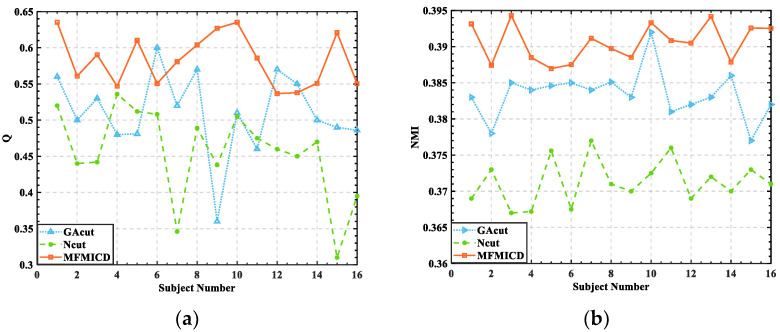
Performance comparison of MFMICD algorithm with Ncut and GAcut algorithms: (**a**) the evaluation function is *Q*; (**b**) the evaluation function is NMI.

**Figure 7 brainsci-12-01159-f007:**
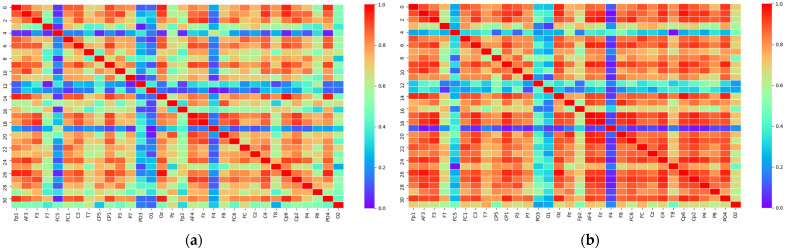
PLV-based fully linked matrix of S2: (**a**) positive status; (**b**) negative status. Different colors represent different weights. The correlation between nodes is inversely correlated with the square’s color: the redder the square, the higher the connection; the bluer the square, the lower the correlation.

**Figure 8 brainsci-12-01159-f008:**
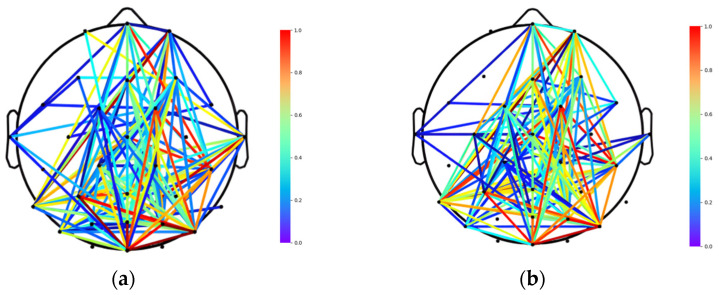
Topology of PLV-based brain network of S2: (**a**) positive status; (**b**) negative status.

**Figure 9 brainsci-12-01159-f009:**
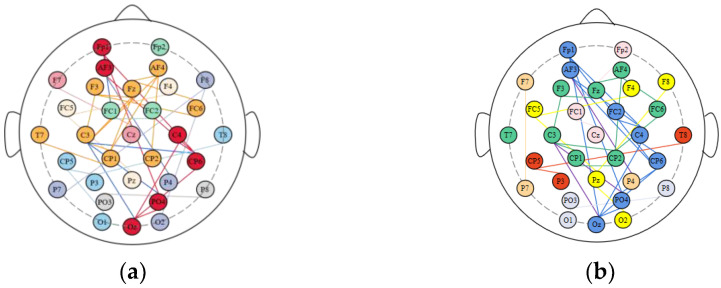
Structural diagram of modular brain network for S2.: (**a**) positive status; (**b**) negative status.

**Table 1 brainsci-12-01159-t001:** Proportion of nodes belonging to each brain region in the top three modules of the S2 brain network to the total number of nodes in each brain region.

		Nodes in Each Brain Region		Percentage	
		Positive State	Negative State	Positive State	Negative State
Frontal Lobe	*Fp1 Fp2 F3 F4 Fz AF3 AF4*	*Fp1 AF3 F3 AF4 Fz F4*	*Fp1 AF3 F3 AF4 Fz F4*	85.70%	85.70%
Temporal Lobe	*F7 T7 P7 FC5 CP5 F8 T8 P8 FC6 CP6*	*CP6 T7 FC6 FC5*	*CP6 T7 FC6 F8 FC5*	40%	50%
Parietal Lobe	*P3 P4 Pz*	None	None	0%	0%
Occipital Lobe	*O1 O2 Oz PO3 PO4*	*Oz PO4*	*Oz PO4 O2*	40%	60%
Central Region	*C3 C4 Cz CP1 CP2 FC1 FC2*	*C4 C3 CP1 CP2*	*C4 FC2 C3 CP1 CP2*	57.10%	71.40%

## Data Availability

The data presented in this study are available upon request from the corresponding author.

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
