# Peer review of "Emotional Brain Network Community Division Study Based on an Improved Immunogenetic Algorithm"

_brainsci, 2022, doi:10.3390/brainsci12091159_

Round 1

Reviewer 1 Report

1. The main drawback of this manuscript is that it has not been presented well enough. It is difficult to understand and follow the explanations. There are several long and unclear sentences, which are difficult to understand the core message. In particular, the Results and Discussion section is very confusing. The authors need to modify and rewrite all parts, especially the Result and Discussion section.

 2. Lines 114-116: “The theta-band data are filtered out for testing after the time series segments in the steady state in the data set are intercepted”. It is not clear what the “testing” means? Are there training and testing steps?

Why do the authors use only the theta band for their analysis? How about other frequency bands? Please, describe it clearly in the manuscript.

 3. Line 138: “The EEG data from the first 32 channels were utilized to perform the studies”. It is not clear what the “first” 32 channel means.

 4. Lines 157-158: “The PLV matrix values that are obtained are all between 0 and 1, with 0 denoting synchronization and 1 desynchronization”.  It seems that this sentence is wrong about PLV. The value 0 reflects the case where there is no phase synchrony, and 1 reflects the phase synchronization.

 5. Lines 334-335: “Individuals in negative emotions have higher module degree values than those in positive emotions, as shown in Figure 4.” Please, explain how the negative emotions have higher module degree values, when the figure shows negative emotions have lower Q values compared to positives.

 6. Line 375: “The individual with the smallest difference in arousal, S2, was chosen…”.

First, the authors need to provide the table to show the subjects’ scores by SAM questionnaire. Then, the reader can clearly observe the subjects’ differences in arousal and other measures, using such table.

Second, it is not clear why the individual with the smallest difference in arousal was chosen. Please, describe it clearly.  

 7. Lines 386-389: “Negative and positive emotions have different topologies of brain networks under the same threshold, with negative emotions having tighter network connections, more weighting of linked edges, and closer information exchange”. Did the authors find these findings from Figs. 7 & 8? How do these figures show “different topologies”? Did the authors perform any statistical comparison and find any significant difference?

 8. Lines 403-407: “From the figure, it can be obtained that: the brain network module in positive emotion has fewer internal nodes and more modules, with a tighter link between each functional module; the brain network module in negative emotion has more internal nodes and less module degree, with a closer connection inside each functional module”.

It is difficult to understand these findings from the figure. It is better to provide a table to show the number of internal nodes and modules for positive and negative emotions. In addition, the authors need to explain how they found “tighter link between each functional module” in positive emotion, and “more internal nodes and less module degree” in negative emotion.  Everything should be clearly explained.

 9. It seems that a majority part of the results is based on the findings on subject S2. Figs. 7, 8, & 9, Table 1, and all the following discussed results are only based on this subject. How do the authors can generalize their findings when they are based on one subject?

 10. English language of the manuscript needs to be improved extensively. Some sentences need to be rephrased and modified. Further, there are many sentences in the manuscript that are very long and unclear and therefore, need to be rewritten in a clearer form. Some unclear sentences include (, but not limited to) these examples:  

 - Lines 21-23: “More brain regions are engaged in emotion dominance in positive states, information exchange between local modules is more frequent, and various emotions cause varied patterns of brain area interaction.”

 - Lines 114-116: “The theta-band data are filtered out for testing after the time series segments in the steady state in the data set are intercepted.”

 - Lines 143-147: “Because the connectivity pattern of the subjects' functional brain networks was difficult to keep relatively stable at the start and end of the data collection phase, the 60s time series was first divided into three equal parts of 20s each before constructing the network, and the middle one (21s , 40s) was finally chosen as the time series for calculation.”

 - Lines 365-369: “The MFMICD algorithm was compared to the Ncut and GAcut algorithms on brain networks in the dataset, where the experimental subjects were EEG data from individuals with subject numbers 1-16 watching 40 movies, and the algorithm was run 50 times independently, with the modularity Q and NMI averaged and the relationship plotted as shown in Figure 6.”

 - Lines 375-379: “The individual with the smallest difference in arousal, S2, was chosen to focus on the brain networks of positive and negative emotions on the premise that the difference between the lowest and highest pleasantness in the individual's 40 experiments was the largest, in order to target and analyze differences in brain networks of the same individual under different emotions.”

 - Lines 383-386: “Figures 7 and 8 show that there is a relatively predictable pattern in the PLV-based all-connectivity matrix, with individuals having higher PLV metric values in the negative state than in the positive state, indicating greater strength of brain network connectivity and higher synchronization between different brain regions.”

 11. Minor comments

- Line 119: “We think that there exist linked edges between …,”. it seems that it is better to rephrase the sentence into: “We assume that there exist linked edges between …”

- There are some abbreviations, which their full forms have not been provided in the first use. For example, the authors used “NMI” several times before they provide the full form in line 278.  

Reviewer 2 Report

I am reviewing “Brain Network Community Division Study Based on an Improved Immunogenetic Algorithm” for Brain Sciences.  As this paper is out of range of my expertise, I cannot contribute to the content of the paper.  I probably should not have agreed to review the paper, but it was on emotions, and I do research and review articles on emotions.  I have many grammar suggestions.

Line 13 on page 1 should say “was suggested for the challenge” and line 14 should spell out BCI before the acronym is used.  Line 14 should also say “to improve convergence and accuracy, the”.  Line 16 should say “The electroencephalographic …database (DEAP) was used to investigate”.  Lines 18-20 should say “The findings revealed that synchronization between…positive condition than in the negative one when four brain areas work together to process emotions.”  Lines 21-23 should say “In positive emotion states, more brain regions…dominance, the information exchange between local modules is more frequent, and various emotions caused more varied patterns of brain area interactions than in negative brain states.”

Line 29 should say “Emotions are the expression of…objective items and they play”.  I replaced “things” with “items”; the authors should not use a vague work like things.  Lines 31-32 should say “Depending on people’s comprehension and evaluation of the situation, they may respond.”  Line 40 should say “neural activity that were generated by various emotions.”  Line 45 should say “researchers because the brain areas”. 

Line 46 on page 2 should say “consideration to thoroughly”.  Line 51 should say “Researchers are currently attempting”.  Line 63 should say “to evaluate the topological”.  Line 66 should say “together.  Therefore, the application”.  Line 77 should say “To solve the problems mentioned above, the scientists”.  Line 78 should spell out “GN” before the acronym is used and acronyms should not be used to start sentences.  Line 80 could follow “Faster Newman” with “(FN)”.  Line 81 should spell out NMF before the acronym is used.  Line 86 should say “and it uses”.  What is modularity Q and NMI?  The authors should explain.  Line 92 should say “proposed to allow” and line 94 should say “different brain network features to increase”.  Line 96 should say “techniques are reliable”.  Line 102 says “Physiological hypotheses” but the authors should describe them.  Lines 104-105 should say “The results and discussion…brain networks are provided in Section 3.”  Lines 114-121 should list the items and use individual sentences to describe them after listing them.  Line 131 should say “We used the Database for Emotion…”  Lines 158-159 should say “obtained all range between”. 

Line 166 should say “Langer et al.”  Line 175 should say “can change dynamically”.  Lines 176-177 should say “making it easy to analyze…various emotional states.”  The authors said “situations”, which refers to the environment, not internal states. 

Lines 213 should say “non-negative to more directly”.  Line 218 should say “Fit(x) to represent”.  Line 219 should say “equation 2” I believe as should the other equations.  Parentheses hide information in sentences.  Line 220 should say “where f(x)”.  Line 226 should say “To preserve”.  Line 233 says “This should be done”.  What should be done?  Line 235 talks about integers between 0 and 1, but integers are whole numbers. Lines 237-238 should say “this work is as stated in equation 3”. 

Line 239 should say “The function fulfills”. Line 244 should say “as stated in equation 2”.  Line 245 should say “equation 3 yields”.  Lines 252-254 should say “High ratios indicated a high probability…individual would be selected.  The population size is assumed to be N”.  Line 262 should say “prevent a circular”. 

Line 279 should say “modularity Q, were used”.  Lines 287-288 should say “High NMI scores suggest that…are comparable”.  Line 289 should say “division, high degree of similarity indicates good algorithm outcomes.”  Line 305 should say “A good community partitioning outcome indicates good network modularity Q values.”

Lines 315-317 should provide the reasons that the numbers were set at certain points.  Line 320 should say “structure, where varied threshold choices greatly impact the final”.  Line 334 should say “Individuals experience negative emotions demonstrated higher…values than individuals experiencing positive emotions”.  Line 336 should say “individuals are experiencing positive emotions.”  Lines 335-336 should say “weak when individuals are experiencing positive emotions.  However, the…values were not significantly different for negative and positive emotions”.  Lines 339-341 should say “individuals are experiencing.  Furthermore, the brain function network demonstrated a certain….structure and it showed the same trend…100, meaning that the value”.  Line 344 should say “This study”.  Line 345 should say “indicating a high”.

Line 350 should say “individuals experiencing negative emotions”.  Line 351 should say “persons experiencing positive emotions”.  Line 352 should say “people are experiencing positive emotions.”  Line 354 should say “showing that the absence of a substantial difference”.  Line 355 should say “various states is not critical, the difference in community structure is important.”  Line 57 should say “NMI to obtain accurate division results”. Lines 362 should say “1) be used to investigate…., 2) assess the brain’s…, and 3) quantify the modular…”.  Line 366 should say “experimental participants’ EEG data came from individuals with participants 1 through 16 watching”. 

Line 373 should say “is resilient and may”.  The second paragraph should be connected to the first paragraph by moving the second paragraph after the first paragraph.  Line 378 should say “largest to target”.  Line 383 should say “show a relatively predictable”.  Line 384 should say “individuals demonstrating”.  Line 395 should say “communities display their own postures”.  Line 396 should say “under different emotions shows communalities”. 

Lines 404-410 should say “The figure shows that…module for positive emotions, which demonstrates fewer……modules with a tighter link between each functional module, than the brain network module for negative emotions, which demonstrates more internal nodes, and a weaker module degree with a closer connection inside each functional module than positive emotions.  The results suggest that …concentrated and demonstrated a high ability”.  Line 415 should say “The findings indicate that”.  Line 416 should say “correspond to four separate”.  Lines 418-420 should say “suggests that each brain region does not work independently when the brain is involved in emotional processing.  Rather, the four brain regions work…..each brain region contains its own”.  Line 423 should say “parietal region does not influence mood alterations…physiological findings”.  A citation should be provided at the end of this sentence.  Line 425 should say “and it is responsible for helping”.  Line 429 should say “To investigate the variability”.  Line 430 should say “largest modules.  The proportion”.  Line 433 should say “brain network accounted”.  Line 435 should say “to the analysis in the above”.  Lines 436-438 should say “negative emotions involved more nodes than with proportions that are greater in the frontal lobe, the occipital love, and central regions than for positive emotions than for negative emotions, which shows that negative emotions affect more brain areas than positive emotions and the information”.         

Line 439 should say “when individuals participate in emotional”.  Line 445 should say “emotions demonstrating more…negative emotions demonstrating more”.  Line 460 should say “processing through” and lines 460-461 should say “and that the brain network connections are stronger and the synchronization between different brain different brain regions is higher during negative states than during positive states.”  Lines 462-464 should say “More brain…dominance, information…are more frequent, and different emotions induce different …interactions for positive emotions than for negative emotions.  The frontal lobe and central”. 

Reviewer 3 Report

The paper presented a novel method for the classification of communities in brain EEG signals during the expression of emotions. While the method presented it interesting and applicable for others in the field, the way in which the paper was written and structured could be improved. The abstract needs to be rewritten to at least tell us what the point of the paper is-the abstract and the title were both too broad. For example the title should perhaps mention "emotions" as this was the main result presented. For the abstract, it will help to structure this a little. The first sentence with "in various phases of BCI research" makes little sense especially since most readers would not know what these various phases should be. Also abrreviations (eg BCI) should be avioded in the abstract. I feel the whole abstract needs to be re-written.

The paragraph from line 65- referencing does not need initals- eg. it should be She et al. not She,Q. et al. Please go through the wole paper to make these changes. Rather than listing who did what- please rewrite this paragraph so that the reader could understand what you are trying to present. 

The paragraph from line 77 containes abreviations that are not explained- what does GN and NMF stand for? And what is Q- given that this is an important concept for your paper.

Rather than main contributions- it will help if you have aims instead. Contributions fits better as conclusions

Why was theta chosen (filtered out for analysis)?  

Even though it is an open dataset- more details on the particpants and study design etc should be presented. 

Line 326 onwards- Why 2 participants? Why not all participants and test whether the differences are significant or not? Line 334- you cant say this as it is based on n=2 only. Also explain what module degree is- it is not Q? Isnt the figure showing positive emotion to have higher q? Results seem to be same (blue on top and red below) for NMI so why is this "in contrast"?

Discussion should include references to other literature to back your finding. 

Round 2

Reviewer 1 Report

The authors made significant changes to the manuscript. They addressed the reviewer’s comments and questions. However, my concern is still the way of presentation as they did not completely address the issue of long and unclear sentences in the manuscript. Some changes and corrections need to be applied to the manuscript.

Some examples:

Lines 410-415 : “To investigate the key brain regions associated with emotional activity, the modular brain network structure of S2 under negative and positive emotions was mapped as shown in Figure 9, with electrodes of different colors belonging to different modules, and inter-node correlations exceeding a set threshold were considered to have connected edges present, where the blue line (left panel) and the purple line (right panel) represent the connections between different modules under the two emotions, respectively.”

Lines 417-421: “The figure shows that the brain network module for positive emotion, which demonstrates fewer internal nodes and more modules, than the brain network module for negative emotions, which demonstrates more internal nodes, and a weaker Q with a closer connection inside each functional module than positive emotions.”

Lines 448-453: “Furthermore, negative emotions involved more nodes than with propor- tions that are greater in the frontal lobe, the occipital lobe, and central regions than for positive emotions than for negative emotions, which shows that negative emotions affect more brain areas than positive emotions and the information connection between local modules is more frequent when individuals participate in emotional dominance.”

Reviewer 3 Report

The authors have adequately addressed my concerns.
